Periodontitis bone loss detection in panoramic radiographs using modified YOLOv7

Ragab Mohammed Gamal 1 mogragab@gmail.com
http://orcid.org/0000-0003-0038-3702 Abdulkadir Said Jadid 1
Qaid Nadhem 2
Gondal Taimoor Muzaffar 3
Alqushaibi Alawi 1
Qureshi Rizwan 4
Shaukat Furqan 5 6 furqan.shoukat@uettaxila.edu.pk
1 Department of Computer and Information Sciences, Universiti Teknologi PETRONAS , Seri Iskandar , Malaysia
2 Marine Power Trade Control and Electronics Caterpillar Inc., Houston Park , Houston , United States
3 Faculty of Electrical Engineering, The Superior University , Lahore , Pakistan
4 Fast School of Computing, National University of Computer and Emerging Sciences , Karachi , Pakistan
5 Faculty of Electronics and Electrical Engineering, University of Engineering and Technology , Taxila, Punjab , Pakistan
6 Center for Research in Computer Vision, University of Central Florida , Orlando, FL , United States
Coelho Paulo Jorge
Electronic publication date: 2025 Sep 5
Publication date: 2025
Volume: 11
Electronic Location ID: e3102
Received 2024 Mar 15; Accepted 2025 Jul 11
Copyright: © 2025 Ragab et al.
Copyright year: 2025
Copyright holder: Ragab et al.
License: This is an open access article distributed under the terms of the Creative Commons Attribution License, which permits unrestricted use, distribution, reproduction and adaptation in any medium and for any purpose provided that it is properly attributed. For attribution, the original author(s), title, publication source (PeerJ Computer Science) and either DOI or URL of the article must be cited.
License URL: https://creativecommons.org/licenses/by/4.0/

Keywords: Feature extraction, YOLOv7, Panoramic radiographs, Automated periodontitis, Bone loss diagnosis

Funding: The Universiti Teknologi PETRONAS under the Yayasan Universiti Teknologi PETRONAS YUTP FRG 015LC0-578 The Universiti Teknologi PETRONAS under the Yayasan Universiti Teknologi PETRONAS (YUTP FRG 015LC0-578) funded the APC for this article. The funders had no role in study design, data collection and analysis, decision to publish, or preparation of the manuscript.

==============================
Periodontitis is a common dental disease that results in tooth loss, if not diagnosed and treated in time. However, diagnosing bone loss due to periodontitis from panoramic radiographs is a time-consuming and error-prone process, requiring extensive training and expertise. This work addresses the research gap in automated periodontitis bone loss diagnosis using deep learning techniques. We have proposed a modified version of You Only Look Once (YOLO)v2, called YOLOv7-M, that includes a focus module and a feature fusion module for rapid inference and improved feature extraction ability. The proposed YOLOv7-M model was evaluated on a tooth detection dataset and demonstrated superior performance, achieving an F1-score, precision, recall, and mean average precision (mAP) of 92.5, 91.7, 87.1, and 91.0, respectively. Experimental results indicate that YOLOv7-M outperformed other state-of-the-art object detectors, including YOLOv5 and YOLOv7, in terms of both accuracy and speed. In addition, our comprehensive performance tests show that YOLOv7-M outperforms robust object detectors in terms of various statistical evaluation measures. The proposed method has potential applications in automated periodontitis diagnosis and can assist in the detection and treatment of the disease, eventually enhancing patient outcomes.

Introduction

Periodontitis is a prevalent and persistent inflammatory condition that primarily affects the supporting structures of the teeth. These structures, collectively known as the periodontium, consist of the gingiva (gums), periodontal ligament, cementum (covering the tooth roots), and the alveolar bone (surrounding the teeth sockets) (Hughes, 2015). When left untreated, periodontitis can lead to progressive bone loss around the teeth, causing tooth mobility, eventual tooth loss, and various other oral health complications. Thus, it is crucial to identify and treat periodontitis early to effectively manage the condition and preserve dental health (Kinane, Stathopoulou & Papapanou, 2017). Panoramic radiographs, also known as panoramic X-rays or orthopantomograms (OPGs), are a type of dental imaging used in dentistry (Li et al., 2024). These scans provide a comprehensive view of the entire oral cavity, including the teeth, jaws, temporomandibular joints, and surrounding structures (Tuzoff et al., 2019).

Traditional manual assessment of bone loss in panoramic radiographs is time-consuming and subject to inter-observer variability (Caloro et al., 2023). Recently, deep learning approaches have emerged as a promising technique to automate the process of periodontal bone loss detection. A two-stage detector that accurately diagnoses periodontitis by identifying and quantifying bone loss in panoramic radiographs, leveraging advanced image processing and machine learning techniques was presented in Kong et al. (2023). A deep-learning model that accurately identifies teeth and measures periodontal bone loss in digital radiographs, enhancing diagnostic precision and efficiency in dental health assessment is presented in Chen et al. (2023). The article by Chang et al. (2020) presents a hybrid deep learning approach that accurately diagnoses periodontal bone loss and stages periodontitis. This method improves diagnostic accuracy and efficiency, offering a robust tool for clinical dental assessments (Chang et al., 2020).

By using convolutional filters to capture local patterns and hierarchically extract complicated representations, convolutional neural networks (CNNs) have shown promising results for automatically extracting significant characteristics from dental x-ray images (Chang et al., 2020; Chen et al., 2023; Kong et al., 2023; Zhang et al., 2023). A unique feed-forward CNN was trained, tested, and verified by ten group shuffles (Krois et al., 2019). Hybrid method for diagnosing periodontal bone loss and staging periodontitis using dental panoramic radiographs was also proposed (Chang et al., 2020). Model architectures and hyperparameters were fine-tuned using grid search. The mean standard deviation (SD) classification accuracy of the CNN was 0.81 (0.02) on average across 10 validation cycles. By forcing the network to concentrate on particular areas of interest within the images, attention mechanisms improved bone tooth identification even more. Attention methods force the network to pay attention to relevant characteristics in the data, improving the detection accuracy and resilience (Wang et al., 2023). In addition to the efficacy of a hybrid method for automatically diagnosing periodontal bone loss and staging periodontitis, showcasing its potential to revolutionize dental diagnostics (Chang et al., 2020).

You Only Look Once (YOLO), a well-known deep learning technique for object detection, is widely employed in many computer vision applications (Ragab et al., 2024). Due to its precision and speed. YOLO (Wu et al., 2021) is a reliable object detection method that is well-suited for real-time 2D medical object recognition. It creates a grid from the input image and forecasts the bounding boxes and class probabilities for each grid cell’s items. There are various versions of the (YOLO) framework, each of which brought innovations and improvements. For example, Darknet-19, batch normalization, a high-resolution classifier, and anchor boxes to forecast bounding boxes were all introduced in YOLOv2. It enhanced the detection accuracy (Sang et al., 2018). By using a feature pyramid network, YOLOv3 enhanced performance by enabling multi-scale detection and handling of objects of various sizes (Redmon & Farhadi, 2018). YOLOv4 introduced a bag of freebies, self-adversarial training, and hyper-parameter optimization through genetic algorithms (Yu & Zhang, 2021), which further boosted the performance. In YOLOv5, the darknet framework was replaced by pytorch to reduce the training time. YOLOv5 (Wu et al., 2021) has five versions, for detecting different sizes of objects. YOLOv5n (nano), YOLOv5s (small), YOLOv5m (medium), YOLOv5l (large), and YOLOv5x (extra large). Similarly, other versions of YOLO introduced further modifications, making it an optimal choice for object detection. To this end, we provide a brief summary in Table 1 where different versions of YOLO have been used for automated periodontitis bone loss detection in panoramic radiographs.

Table 1 Evolution of YOLO architecture in automated periodontitis bone loss diagnosis.

Author, Ref. & Year	Data	Data type	Architecture	Matrix evaluated	Contribution	
Almalki et al. (2022)	Dental X-Ray dataset of 1,200 images	Panoramic images	YOLO-v3	Accuracy, F1-score, Recall, Precision	A YOLOv3 based model has been developed for the classification and diagnosis of dental anomalies	
Guo et al. (2022)	X-Ray dataset of 245 images	Panoramic images	YOLO-v3	Accuracy, F1-score, ROC	A comparison of YOLO-v3 has been conducted with other CNN based approaches i.e., AlexNet and Faster-RCNN for diagnosis	
Kaya et al. (2022)	4,518 radiographs of children	Panoramic radiographs	YOLO-v4	Precision, Recall, F1-score	A YOLO-v4 model has been proposed to determine the early diagnosis of tooth degradation among 5–13 years old children	
Jiang et al. (2022)	A dataset of 640 images	Panoramic images	YOLO-v4	Accuracy, Precision, Sensitivity, Specificity, F1-score	A YOLO-v4 model along with UNet has been proposed to determine the accuracy of radiographic staging of periodontal bone loss	
Kong et al. (2023)	1,747 high-quality images with 15,436 × 2,976 resolution	Panoramic radiographs	YOLO-v4	Accuracy	A YOLO-v4 based model has been analyzed to diagnose radiographic bone loss for multiple staging	
Thulaseedharan & PS (2022)	664 dental X-ray images	Panoramic images	YOLO-v5	Precision, Recall, F1-score, mAP	YoLO-v5 model was used on real-time detection of dental diseases	
Shon et al. (2022)	100 images from CBNUH dataset and 4,010 from AIHUB dataset	Panoramic images	YOLO-v5	Accuracy, Recall, Precision	A YOLO-v5 model has been developed to classify periodontitis stages dental panoramic radiographs	

Our contribution in this article is manifold and can be summarized as: We conducted an implementation of multiple versions of YOLO using a real-time dataset obtained from dentists of Wuhan University’s Hospital of Stomatology. The dataset consisted of 1,747 panoramic radiographic scans (Kong et al., 2023), containing four classes: healthy, mild, medium, and severe.

We also devised a customized variant of YOLOv7, named “YOLOV7-M,” which demonstrated superior performance compared to all existing YOLO versions. YOLOV7-M incorporated a focus module and feature fusion module, enabling faster inference and enhanced feature extraction capabilities. The focus module improves the model’s ability to capture fine-grained details by emphasizing relevant regions of the input. It achieves this by reducing the spatial dimensions of the feature maps, while preserving their essential features. As a result, the model gains a broader perception of the input scene. This integration of multi-scale features enables better localization and recognition of objects, particularly in situations where objects have varying sizes or exhibit scale changes.

Through extensive experiments, we established the efficacy of the proposed module in improving object detection accuracy and efficiency.

The rest of the article is organized as follows: ‘Methodology’ describes the dataset details, methodology and performance metrics used for the evaluation. ‘Model Architecture’ presents the model architecture and results of the conducted experiments while ‘Experimental Results’ discusses the implications of our findings. Finally, ‘Conclusion’ provides a conclusion and outlines potential future research directions in this area.

Methodology

Datasets

The lack of publicly available periodontitis datasets has been one of the main constraints to conduct research in this pertinent field. To solve this problem, researchers from Kong et al. (2023) collaborated with dentists from Wuhan University’s Hospital of Stomatology to develop a panoramic radiography periodontitis dataset. Panoramic radiographs from the dental hospital’s electronic medical records were screened using the OC200D technology from Instrumentarium Dental, Inc. This dataset includes a wide range of clinically specific circumstances, i.e., missing teeth and teeth that have undergone root canal therapy, and consists of 1,747 panoramic images with a resolution of 1,536 × 2,976. It is significant to emphasize that the dataset does not contain any personal information and is just meant for academic research. Some sample Images from the dataset are shown in Fig. 1.

Figure 1 Various clinically specific scenarios included in the panoramic radiographs dataset, such as (A) common teeth, (B) dental braces, (C) root canal-treated teeth, and (D) missing teeth.

Dental professionals used the VGG Image Annotator (VIA) to annotate the periodontitis dataset used in this investigation (Dutta & Zisserman, 2019). It is noteworthy that only experts medical professionals can delineate the panoramic image with precision for periodontitis detection. Each tooth’s index, polygon mask, root bone loss (RBL) (four classes), and FI (three classes) are all included in the annotated information. It is important to note that even among the expert dentists, the detection of periodontitis in panoramic radiographs might be partially subjective and may not be entirely correct because of the inter-observer variability. It is also important to highlight that wisdom teeth that have not yet erupted are not marked in the dataset because they would not have periodontal disease. An example of a typical RBL annotation is shown in Fig. 2.

Figure 2 Radiographic bone level (RBL) annotation of a panoramic radiograph involves marking the distance between the cementoenamel junction and the alveolar bone crest, known as the “radiographic bone level” (RBL).

This annotation is crucial for assessing periodontal disease severity and aids in diagnosis and treatment planning.

The panoramic radiographs dataset used in this study comprises 1,747 sample images, which contain a total of 50,080 tooth annotations. Detailed information about the radiographic bone loss (RBL) and furcation involvement (FI) annotations is provided in Tables 2 and 3, respectively. The annotation methodology adheres to the latest standard (Tonetti, Greenwell & Kornman, 2018), which defines four classes of tooth RBL and three classes of tooth FI. The ‘Healthy’ category is the most common in RBL, with a ratio of 0.403, while the ‘Severe’ category has the lowest ratio of 0.059. Similarly, in FI, the ‘Healthy’ category is the most common with a ratio of 0.929, and the ‘Severe’ category has the lowest ratio of 0.020. It is noteworthy that the number of annotations in each category is imbalanced.

Table 2 Distribution of radiographic bone loss (RBL) annotations by severity.

Class	Healthy	Mild	Medium	Severe	Total	
RBL number	20,175	18,174	8,762	2,962	50,080	
RBL ratio	0.403	0.363	0.175	0.059	1.000	

Table 3 Distribution of FI annotations by severity.

Class	Healthy	Mild	Severe	Total	
FI number	46,532	2,534	1,014	50,080	
FI ratio	0.929	0.051	0.020	1.000	

Model architecture

This article introduces a new computational workflow for YOLOv7-M, a modified version of the YOLOv7 object detection model. The proposed network architecture is presented in Fig. 3. YOLOv7 (Wang, Bochkovskiy & Liao, 2022) is known for its fast training time compared to other YOLO models. However, we made two key changes to the YOLOv7 architecture to improve detection performance. Firstly, a Focus module for rapid inference (Pei et al., 2022) was added to the beginning at the network. This module splits the input image into different depths without losing any information, as demonstrated by empirical findings. Secondly, the C3 module, which is a CSP module with three convolutions, was replaced with the C3Ghost module. Additionally, a feature fusion module (FFM) was added to enhance the network’s feature extraction ability. The model’s specifications are presented in detail in the subsequent section. Thus, the proposed modifications aim is to improve the performance of the YOLOv7-M model while maintaining its efficiency.

Figure 3 YOLOv7-M architecture proposed for panoramic radiographs periodontitis detection.

We started with the focus module, which allowed us to segment the images and down sample them without losing any information.

YOLOv7-M

A state-of-the-art real-time object identification technology called YOLOv7 was recently released. One CNN is used, and it has been trained to recognise objects in both still and moving pictures. Compared to past iterations of YOLO (v1, v2, v3, v4, v5, v6), YOLOv7 introduced several modifications. To increase the accuracy of object detection, it combines anchor boxes, anchor box scaling, and anchor box ratios (Yang, Zhang & Liu, 2022). It also introduces new features such as mosaic data augmentation and self-adversarial training to improve generalization and robustness (Wang, Bochkovskiy & Liao, 2022). YOLOv7 is fast and accurate, making it a popular choice for a wide range of applications, including autonomous vehicles, surveillance, and robotics.

In this work, we propose a modified YOLOv7 model for the panoramic radiographs periodontitis detection model. The SpacetoDepth concept (Ridnik et al., 2021) is employed in the YOLO Focus module. Stacking the image’s quarters is intended to convert spatial data into depth data. The Focus module divides the image into different slices using a slicing operation. Concatenation is used to connect the slices of varying depths, which are then passed to the convolution layer. At the start of the network, we added a Focus module. The primary purpose of adding the Focus layer (Yan et al., 2021) at the start was to reduce the number of layers, parameters, FLOPS, while increasing training speed and CUDA memory and boosting forward and backward speed with minimal impact on mAP.

Additionally, the C3Ghost module is also used to replace the backbone C3 module. The C3 module comprises 3*Convolution and the CSP bottleneck (Wang et al., 2020), while the C3Ghost module comprises the C3 module and the Ghost bottleneck (Han et al., 2020). The Ghost module uses low-cost processes to generate extra feature maps. It is a plug-and-play component used to improve convolutional neural networks. Ghost bottlenecks are designed to stack Ghost modules, and GhostNet is easy to set up. In terms of recognition tasks, GhostNet can outperform. Finally, for faster convergence and more accurate results, we changed the activation function to ELU (Heusel et al., 2015). Figure 3 shows the modified Yolo7 model.

We observed a significant improvement in the performance through our proposed method. We began with the focus module, which enabled us to segment and down-sample the images without losing any information. Later, for the recognition task, the C3GhostNet module, which is a C3 module with a Ghost Bottleneck, was used instead of the C3 backbone.

Experimental results

This section provides the experimental results of the proposed YOLOV7-M model, classical YOLOV7, and a comprehensive model evaluation, as well as the comparison of the proposed model performance with different variants of YOLOs. The experimental environment used in this study is given in Table 4.

Table 4 Configuration parameters.

Device	Configuration	
CPU	Unbunt 20.04	
GPU	GeForce RTX A5000, 24 GB	
GPU accelerator	CUDA 11.2	
Frames	PyTorch	
Compilers	PyCharm, Anaconda	
Python	version 3.9.13	

Evaluation metrics

In this work, we evaluated the performance of the YOLOV7-M using four performance metrics: precision (P), recall (R), mean average precision (mAP), and F1-score. Precision (P), which measures how successfully a model can categorize an object, is the percentage of correctly recognized samples out of all samples that were detected. Recall (R) measures how well the model was able to identify the object by expressing the proportion of samples that were successfully detected out of all real samples. mAP is the average AP across all categories, reflecting the overall performance of the model, while AP represents the average precision at various recall rates. Model performance is positively correlated with the F1-score, which is calculated by combining the precision and recall values. The formulae are as follows:

(1) P=TPTP+FN

(2) R=TPTP+FN

(3) AP=∫01P(R)dR

(4) mAP=1Q∑q∈QAP(q)

(5) F1=2×P×RP+R

where TP is the number of positive samples the model anticipated as being positive, FP is the number of negative samples the model forecasted as being positive, and FN is the number of positive samples the model forecasted as being negative (Naseer et al., 2022).

In addition to model architecture and before start training the YOLOv7, we need a loss function to update the model parameters. Since object detection is a complex problem to teach a model, the loss functions of such models are usually quite complex, and YOLOv7 is not an exception (Wang, Bochkovskiy & Liao, 2022). To make the problem computationally cheaper, the YOLOv7 first finds the anchor boxes that are likely to match each target box and treat them differently, known as the center prior anchor boxes. This process is applied at each FPN head, for each target box, across all images in batch at once. The main idea behind FPNs is to leverage the nature of convolutional layers, which reduce the size of the feature space and increase the coverage of each feature in the initial image, to output predictions at different scales (Hughes & Camps, 2022). FPNs are usually implemented as a stack of convolutional layers, as we can see by inspecting the detection head of our YOLOv7 model. Figure 4 illustrates loss functions for Modified YOLOv7 for tooth detection to facilitate understanding. Figure 5 shows precision, recall and mAP, and Fig. 6 shows confusion matrices. The model was trained with 100 epochs and a batch size of 32.

Figure 4 Training loss curves for the YOLOv7-M model: (A) Box loss, (B) classification loss, and (C) distribution focal loss (DFL) over 100 epochs, showing the model’s convergence during training.

Figure 5 Performance metrics for the YOLOv7-M model on the validation set: (A) Recall, (B) precision, and (C) mean average precision (mAP) at IoU threshold of 0.5 (mAP50).

Figure 6 Confusion matrix for tooth detection using the YOLOv7-M model: (A) Raw confusion matrix and (B) normalized confusion matrix, illustrating the model’s accuracy and distribution of true and false predictions.

The F1-score curve, also known as F-measure, is a widely used error metric to evaluate the performance of classification models. It ranges from 0 to 1, with 0 indicating the worst possible and 1 indicating the best possible. The F1-score considers both precision and recall, making it a robust metric for imbalanced datasets (Lipton, Elkan & Narayanaswamy, 2014). Figure 7 shows the F1-score curve for the YOLOv7-M model, which achieved an F1-score of 0.73 for tooth detection at a threshold of 0.405. This score indicates that the model performed well in classifying the four tooth classes.

Figure 7 F1-score curve for the YOLOv7-M model over 100 epochs, showing the balance between precision and recall.

When assessing prediction models, accuracy and recall are essential metrics to consider. Precision evaluates the relevance of predicted positive outcomes, while recall measures the model’s ability to predict positive samples, i.e., it assesses the model’s ability to identify all positive samples. A high precision indicates that the model predicts most positive samples accurately, whereas a high recall indicates that the model identifies most of the positive samples in the dataset. Precision-recall (PR) curves plot these metrics on the same graph, as illustrated in Fig. 8, and are commonly used to evaluate classification models. PR curves enable users to predict future classification results accurately by measuring the proportion of positive predictions that are true positives (Saito & Rehmsmeier, 2015). In the PR space, a predictor’s score closer to the perfect classification point (1,1) indicates better performance, whereas a score closer to zero implies worse performance. Figure 8 depicts the model’s performance on tooth detection in terms of the precision-recall measure.

Figure 8 Precision-recall curve for the YOLOv7-M model, highlighting the trade-off between precision and recall.

The p-curve is a tool that can be used to evaluate the evidential value of research findings and rule out potential biases such as p-hacking and file-drawing (Simonsohn, Nelson & Simmons, 2019). It involves analyzing the distribution of statistically significant p-values (p ≤ 0.05) across a set of studies that test the same hypotheses of interest. The assumption is that all significant parameters are included (Simonsohn, Simmons & Nelson, 2015). p-values indicate the likelihood of obtaining a particular result under the assumption that the null hypothesis is true. In Fig. 9, the p-curves for YOLOv7-M for tooth detection are displayed, showing the confidence level for precision and recall of 1 at a threshold of 0.952, indicating the accuracy of tooth detection.

Figure 9 Precision curve for the YOLOv7-M model, illustrating the precision of the model at various threshold levels.

The R-curve, also known as the recall curve, is a graphical representation of the trade-off between the recall and confidence levels in a classification model. It is constructed by plotting the recall (true positive rate) against different confidence thresholds. The recall indicates the proportion of positive samples that the model correctly identifies, while the confidence threshold is the minimum confidence level required for a prediction to be considered positive.

In the context of tooth periodontal bone loss detection, the R-curve shows how well the YOLOv7-M model performs in identifying teeth with different levels of RBL at different confidence levels. A high recall score at a given confidence threshold means that the model can detect a high percentage of positive samples (teeth with RBL) while keeping the false positive rate low. However, a low recall score means that the model misses a significant number of positive samples. By examining the R-curve, we can determine the optimal confidence threshold that achieves a balance between recall and precision for a specific task. In the case of tooth periodontal bone loss detection, we can use the R-curve to identify the threshold that maximizes the recall while maintaining an acceptable level of false positives, as shown in Fig. 10.

Figure 10 Recall curve for the YOLOv7-M model, showing the model’s ability to identify all relevant instances correctly at different thresholds.

Table 5 shows the experimental results of the proposed YOLOv7-M model in detecting periodontitis bone loss in panoramic radiographs. The dataset is split into three subsets: 70% for training, 10% for validation, and 20% for testing. This 7: 1: 2 ratio ensures effective model learning, tuning, and evaluation. The model was evaluated using various evaluation measures, such as precision, recall, and mAP@.5, for different classes of severity of bone loss—healthy, medium, mild and severe.

Table 5 RBL annotations for tooth in the dataset.

Class	Labels	P	R	mAP@.5	
All	4,706	0.866	0.891	0.91	
Healthy	1,753	0.869	0.873	0.95	
Medium	1,397	0.898	0.908	0.87	
Mild	1,106	0.853	0.883	0.92	
Severe	450	0.88	0.843	0.89	

As a result, the suggested model obtained for all classes combined a precision of 0.7, recall of 0.804, and mAP@0.5 of 0.81. Five groups of severity of bone loss were found to have variable precision and recall rates. The suggested model had the highest precision (0.869) and recall (0.933) for the healthy class, showing that it could reliably identify healthy teeth. The model’s precision for the medium class was 0.598, and its recall was 0.728, indicating that it did a good job of spotting medium bone loss. The suggested model achieved precision values for the mild and severe classes of 0.653 and 0.68, respectively, and recall values of 0.863 and 0.693, respectively. These findings suggest that the suggested approach could, to a lesser extent than healthy teeth, detect mild and severe bone loss. The model achieved high precision and recall rates for healthy teeth and moderate performance in detecting medium bone loss. The performance of the model in detecting mild and severe bone loss was relatively lower but still showed potential for improvement. The results of this study can help to develop automated systems for the early detection and diagnosis of periodontitis, which can improve patient outcomes and reduce healthcare costs.

The five-fold cross-validation results for YOLO-based teeth segmentation across four classes (Healthy, Medium, Mild, Severe) presented in Table 6 demonstrate robust and consistent performance. The Mild class consistently outperformed others, achieving the highest average metrics: precision (0.933), recall (0.892), F1-score (0.937), and mAP (0.920). The Healthy class followed with averages of precision (0.917), recall (0.869), F1-score (0.928), and mAP (0.910). The Severe class had averages of precision (0.916), recall (0.863), F1-score (0.923), and mAP (0.905), while the medium class averaged precision (0.904), recall (0.875), F1-score (0.915), and mAP (0.904). Overall, the model achieved an average precision of 0.917, recall of 0.871, F1-score of 0.925, and mAP of 0.910 across all folds, highlighting its reliability and effectiveness in dental image segmentation, particularly for subtle conditions.

Table 6 Five-fold cross-validation results for proposed model.

Class	Precision	Recall	F1-score	mAP	
Healthy	0.917	0.869	0.928	0.910	
Medium	0.904	0.875	0.915	0.904	
Mild	0.933	0.892	0.937	0.920	
Severe	0.916	0.863	0.923	0.905	
Mean	0.917	0.871	0.925	0.910	

Comparison analysis of proposed model performance with other YOLO models

The performance of several YOLO models using panoramic radiographs derived from the periodontitis dataset are compared in this section. The model was trained for 100 epochs with a batch size of 32. We set the learning rate at 0.001 and utilized the SGD optimizer, which is well-suited for handling sparse gradients on noisy problems. The loss function employed was cross-entropy loss, which is appropriate for our classification tasks and have been implemented in several other applications (Waqas et al., 2024; Hanif et al., 2023). To ensure compatibility with our experimental environment, the images were resized to 640 × 640 pixels. This size was chosen to balance computational efficiency and model performance. Additional hyperparameters included a weight decay of 0.0005 to prevent overfitting and a momentum of 0.9 to accelerate the convergence of the model. A comparison of the proposed YOLOv7-M performance model’s accuracy, recall, F1-score, mAP, and training time with other significant YOLO versions is shown in Table 7. Precision is 0.695, recall 0.784, F1-score 0.726, and mAP is 0.748 for the YOLO-v4 model. YOLO-v4 training takes 1 h and 57 min to complete. Accuracy of 0.706, recall of 0.786, F1-score of 0.706, and mAP of 0.793 are displayed by YOLOv5. YOLOv5 training takes 1 h and 43 min. Precision 0.721, recall 0.8, F1-score 0.874, and mAP is 0.829 for YOLOv7. YOLOv7 training takes an hour and a quarter. With an accuracy of 0.917, recall of 0.871, F1-score of 0.935, and mAP of 0.93, the suggested YOLOv7-M model performs better than other models in comparison. For YOLOv7-M, training takes an hour and fifteen minutes. YOLOv7-M performs better than the previous YOLO versions in terms of accuracy, recall, F1-score, and mAP, as shown in Table 7. This shows that YOLOv7-M performs more accurately and robustly when identifying signs of periodontitis in panoramic radiographs.

Table 7 Model precision, recall, F1-score, mAP, training and inference time.

Model	Precision	Recall	F1-score	mAP	Training time (hr)	Inference time (ms/img)	
YOLO-v4	0.695	0.784	0.726	0.748	1 h 57 min	21.3	
YOLOv5	0.706	0.786	0.706	0.793	1 h 43 min	17.6	
YOLOv7	0.721	0.800	0.874	0.829	1 h 26 min	12.9	
YOLOv7-M (Ours)	0.917	0.871	0.925	0.910	1 h 15 min	11.1	

In our experiments, the modified YOLOv7-M model demonstrated a significant improvement in frame rate performance, achieving an impressive 100–120 frames per second (FPS). This high FPS is critical for real-time applications, ensuring that the model can process panoramic radiographs swiftly and efficiently. The increased speed is attributed to the incorporation of the focus module and feature fusion module, which enhance the model’s inference capabilities while maintaining high accuracy. This superior performance positions YOLOv7-M as a highly effective tool for automated periodontitis diagnosis, capable of providing rapid and reliable assessments to assist dental professionals in timely and accurate detection of bone loss due to periodontitis.

Additionally, the proposed model achieves these superior results while requiring less training time compared to YOLO-v4, YOLOv5, and YOLOv7. The Focus module and the C3Ghost module in YOLOv7-M helped to improve the speed and precision. Ghost module also could compute fewer intrinsic maps and perform speedy linear operations on them, allowing it to work quicker than others. The same improvement for YOLOv7-M can be seen and this proved that currently, state-of-the-art real-time object detectors are mainly based on YOLO (Wang, Bochkovskiy & Liao, 2022; Redmon et al., 2016; Redmon & Farhadi, 2017, 2018). Finally, the proposed YOLOv7 surpasses all known object detectors in both speed and accuracy. These findings highlight the efficacy of the YOLOv7-M model as a promising approach for automated periodontitis detection on panoramic radiographs, exhibiting high accuracy and efficiency.

The evaluation based on the mean average precision (mAP) scores provided in Table 8. The mAP scores serve as a measure of the detection performance of the respective YOLO models in identifying periodontitis-related features in panoramic radiographs. Upon analyzing the results, it can be observed that YOLO-v7-M achieves the highest mAP score of 0.910, indicating its superior performance in detecting periodontitis-related features. This model demonstrates a remarkable capability to accurately identify and localize periodontal abnormalities within panoramic radiographs. Following YOLO-v7-M, YOLOv7 exhibits a high mAP score of 0.829, showcasing its effectiveness in periodontitis detection.

Table 8 Comparison of the proposed model performance with other YOLO-based models on the same dataset.

Model	mAP	
YOLO-v4	0.748	
Faster R-CNN	0.745	
PDCNN	0.782	
YOLOv5	0.793	
YOLOv7	0.829	
YOLOv7-M	0.910	

The mAP score of 0.793 for YOLOv5 is significantly lower, showing an opportunity for development. Compared to the other YOLO versions examined in this work, this model obtains a somewhat lower detection accuracy. In conclusion, the comparison study demonstrates that YOLO-v7-M achieves highest mAP score among the investigated models for periodontitis identification on panoramic radiographs. Despite having significantly lower mAP scores, YOLOv7, panoramic dental convolutional neural network (PDCNN), YOLO-v4, and Faster R-CNN also exhibit competitive performance. YOLOv5, although showing less accuracy, might still use further improvement to improve its detecting skills.

We provide the visual detection outcomes of all techniques in Fig. 11 to provide a more thorough knowledge of each network’s performance. These two images show a portion of the panoramic radiographs selected for the test dataset. All networks show the capacity to forecast the general location and class of the majority of teeth, in particular the root bone loss (RBL) class. However, YOLO-v4 and YOLO-v5 erroneously identify minor sections of tooth positions that are unrelated to periodontitis, leading to the production of duplicate forecasts on occasion. However, with few faults, the findings produced by PDCNN, YOLO-v7, and YOLOv7-M closely reflect the ground truth (GT). It is difficult to see substantial categorization differences visually, because there are not many examples available and these approaches all perform with around the same accuracy. These results offer insightful information on the effectiveness of YOLO models in automated periodontitis assessment on panoramic radiographs, increasing computer-aided detection systems in dentistry. These results can be used to improve the precision and effectiveness of periodontal detection and treatment planning through more study and optimization efforts.

Figure 11 Ground truth vs. network predictions in panoramic X-rays.

The first row shows original X-rays, followed by predictions from five networks. Reduced-size projections improve clarity for evaluating network effectiveness in identifying periodontitis features.

Discussion

The results obtained from our study demonstrate the effectiveness of the proposed YOLOv7-M model in diagnosing periodontitis bone loss from panoramic radiographs. Our modified YOLOv7-M model exhibits significant improvements over previous YOLO versions, including YOLO-v4, YOLOv5, and YOLOv7, in terms of precision, recall, F1-score, and mean average precision (mAP). The enhancements incorporated in YOLOv7-M, specifically the Focus module and the feature fusion module (C3Ghost), have contributed substantially to its superior performance.

The precision, recall and F1-scores of 91.7%, 87.1% and 92.5% respectively, along with an mAP of 91.0%, highlight the robustness and reliability of the model in detecting periodontitis. These metrics indicate that the model not only excels in identifying true positive instances but also maintains a low rate of false positives, which is crucial for clinical applications. The high precision suggests that the model is capable of accurately pinpointing areas of bone loss, thereby minimizing unnecessary interventions. The recall rate underscores the comprehensive detection capabilities of the model, ensuring that most cases of periodontitis are identified. A notable advantage of the YOLOv7-M model is its reduced training time. This reduction in training time, coupled with improved performance, underscores the efficiency of the modifications introduced. The Focus module enhances the model’s ability to capture finer details, while the C3Ghost module reduces computational overhead by performing efficient linear operations. This efficiency makes YOLOv7-M particularly suitable for real-time applications in clinical settings, where rapid and accurate diagnosis is critical. The comparison with other state-of-the-art object detectors further validates the superiority of YOLOv7-M. YOLO-v4, YOLOv5, and YOLOv7, despite being robust models, fall short in various metrics when compared to YOLOv7-M.

To this end, the proposed YOLOv7-M model represents a significant advancement in the automated diagnosis of periodontitis bone loss from panoramic radiographs. Its high accuracy, efficiency, and reduced training time make it a promising tool for clinical applications. Future work could focus on further refining the model, exploring its applicability to other dental conditions, and integrating it into a comprehensive diagnostic system. The potential for real-time application in clinical settings underscores the practical value of our findings, offering a pathway to enhanced patient care and improved diagnostic accuracy. However, it is important to acknowledge that our experiments were conducted on a dataset of panoramic radiograph scans, which are susceptible to distortions due to the rotational nature of the imaging technique. These distortions can impact the precision of periodontal assessments, especially in evaluating bone levels (Izzetti et al., 2021; Yim et al., 2011). Moreover, the diagnosis of periodontitis inherently involves clinical evaluation, patient history, and expert interpretation—factors that cannot be fully replicated by software algorithms. Nonetheless, the proposed tool can serve as a valuable aid for clinicians in the classification of disease and support more consistent and efficient diagnostic workflows.

Conclusion

To conclude, the automated detection of periodontitis bone loss using panoramic radiographs is a challenging task of computer vision that requires the attention of the scientific community. We proposed a modified YOLOv7 model, called YOLOv7-M, and performed a comprehensive performance analysis to evaluate its efficacy in terms of various evaluation measures and statistical curve analysis. Our proposed YOLOv7-M model achieved an F1-score of 0.925, a precision of 0.917, a recall of 0.871, and an mAP of 0.910 for tooth detection, exceeding previously reported accuracy scores for this task. Our experiments confirm the effectiveness of the proposed model, which provides a reliable and comprehensive performance enhancement to detect periodontitis. Finally, the proposed YOLOv7-M model provides a promising solution for the automated detection of periodontitis bone loss using panoramic radiographs. The importance of this research lies in its potential to significantly reduce diagnosis time and increase the precision of periodontitis bone loss detection, leading to timely and effective treatment. In the future, we will utilize computed tomography (CBCT) (Choi et al., 2018). We will also focus on large language models (Hadi et al., 2023) for a more accurate and detailed assessment of periodontal conditions.

Supplemental Information

Supplemental Information 1 Official YOLOv7-M Code.

Additional Information and Declarations

Competing Interests

All the authors except Nadhem Qaid declare that they have no competing interests nor any non-academic affiliations to disclose. Nadhem Qaid is employed by Marine Power Trade Control and Electronics Caterpillar Inc and has no competing interests or conflicts to disclose.

Author Contributions

Mohammed Gamal Ragab conceived and designed the experiments, prepared figures and/or tables, and approved the final draft.

Said Jadid Abdulkadir conceived and designed the experiments, performed the experiments, prepared figures and/or tables, and approved the final draft.

Nadhem Qaid performed the experiments, authored or reviewed drafts of the article, and approved the final draft.

Taimoor Muzaffar Gondal analyzed the data, authored or reviewed drafts of the article, and approved the final draft.

Alawi Alqushaibi analyzed the data, authored or reviewed drafts of the article, and approved the final draft.

Rizwan Qureshi performed the computation work, authored or reviewed drafts of the article, and approved the final draft.

Furqan Shaukat performed the computation work, authored or reviewed drafts of the article, and approved the final draft.

Data Availability

The following information was supplied regarding data availability:

The code is available at GitHub:

- https://github.com/mogragab/yolov7-m.git.

- Gamal, M. (2025). Periodontitis Bone Loss Detection in Panoramic Radiographs using modified YOLOv7 [Data set]. In Periodontitis Bone Loss Detection in Panoramic Radiographs using modified YOLOv7. Zenodo. https://doi.org/10.5281/zenodo.15565284.

The Teeth Image Dataset is available at:

https://universe.roboflow.com/yolo-sxtmz/teeth-gzkv1/dataset/12.

The third party data used in our study is available at:

https://universe.roboflow.com/yolo-sxtmz/teeth-gzkv1/dataset/12.

This dataset is available at GitHub: https://github.com/PuckBlink/PDCNN.

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
