# Peer review of "Periodontitis bone loss detection in panoramic radiographs using modified YOLOv7"

_PeerJ Computer Science, doi:10.7717/peerj-cs.3102_

## Round 0.1 · original submission · Major Revisions

The authors shall work on clarifying the experimental design and the validity of the findings as recommended by the reviewers and perform a significance analysis of the reported results.

Reviewer 1 ·

Basic reporting

This study introduces a modified version of YOLOv7, YOLOv7-M, which incorporates a focus module and a feature fusion module designed for rapid inference and enhanced feature extraction capabilities. The YOLOv7-M model was assessed using a tooth detection dataset and demonstrated exceptional performance with an F1 score, precision, recall, and mAP of 93.5, 91.7, 87.1, and 93.0, respectively. The method shows promise for automated periodontitis diagnosis.

However, the paper requires substantial revisions and rewriting. Here is a summary of my concerns:
1. While the modification of YOLOv7 to YOLOv7-M is a significant contribution, the application explored in the paper is not novel and has been explored previously. To validate the efficacy of YOLOv7-M, the authors should test it on additional datasets beyond the one used as this is the main contribution of the work. Whenever there is a new network proposed, we expect comprehensive evaluation of the network with >1 datasets.
2. The authors should conduct an ablation study for the newly added modules—the focus module and the feature fusion module. It is crucial to detail the individual contributions of these modules to the overall performance of YOLOv7. Understanding the impact of each module will help validate their effectiveness and necessity in the proposed model. This study should include results with and without each module, along with a discussion of the observed differences. Currently, the discussion of the results is weak. Maybe add a Discussion section to discuss the results and clinical implications of the proposed method.
3. The related work section is insufficient and missed several references. The authors should cite relevant studies such as:
https://www.nature.com/articles/s41598-020-64509-z
https://www.sciencedirect.com/science/article/pii/S1991790223000946
https://www.sciencedirect.com/science/article/abs/pii/S0010482522010824

Comparisons with at least a few existing methods should be included to contextualize YOLOv7-M’s performance. There is a general need for a more thorough engagement with previous literature and for the inclusion of comparative analyses with previously proposed methods.
4. The authors claim that manual assessment of panoramic radiographs is time-consuming and prone to errors due to inter-observer variability. This claims requires evidence through comparative analyses of human and machine processing times, supported by research that includes inter- and intra-observer variability studies. You can cite previous works that proved this claim or you can provide such deviance through inter- and intra- agreement assessment, and time analysis.
5. Reporting the deployment speed of YOLOv7-M in terms such as frames per second (FPS) is needed, as training times are less relevant from a clinical perspective once the model is developed.
6. Did the authors investigate any data augmentation techniques to enhance the diversity of the dataset?
7. The performance should be reported across different demographic groups and specific clinical conditions to avoid biased results.
8. The availability of the dataset and the reproducibility of the results are critical. The authors should state whether the dataset and the code are publicly accessible, and if so, how they can be obtained.
9. The manuscript requires significant improvements in clarity and conciseness. The current draft contains lengthy, awkwardly constructed sentences, and several grammatical and punctuation errors that need correction. Specific issues such as the lack of space in line 52 and the unexplained acronym 'RBL' appearing for the first time in line 103 must be addressed.
10. The quality of the figures presented in the manuscript is not acceptable; it needs enhancement to meet publication requirements.

Experimental design

As above

Validity of the findings

As above

Additional comments

As above

Annotated reviews are not available for download in order to protect the identity of reviewers who chose to remain anonymous.
Cite this review as

Reviewer 2 ·

Basic reporting

The research, titled "Periodontitis Bone Loss Detection in Panoramic Radiographs Using Modified YOLOv7," utilizes open-source data to detect periodontal health status and classify teeth with mild, moderate, and severe periodontitis. The proposed detection method involves a modified version of YOLOv7, as well as the implementation of other YOLO series architectures. However, upon reviewing the paper, the following inconsistency was identified.
1. Could you refine the titles of Tables 2 and 3 to provide a more concise overview, as they are not mentioned in the description? Mentioning tables in line 110 ("Detailed information about the RBL and the FI is provided in Tables 1 and 2, respectively.") is inappropriate.

2. Please ensure that abbreviations are defined at their first mention and used consistently thereafter. Verify the definitions of RBL and FI terms. Check line 264 for PDCNN and other cases as well.

3. Please enhance the image quality, as the text in the images is hard to read. When multiple pictures are grouped together, it's preferable to mark each individual image with roman letters. Additionally, the captions of Figures 2 and 11 are too large; it would be beneficial to provide a concise caption for each figure and explain its features in the description section. Please refine the captions of the other images as well. In Figure 11, you have showcased the outcomes of different models. Could you please identify the fault cases as well? Figures 1 and 2 appear to be copied from "Kong, Z., Ouyang, H., Cao, Y., Huang, T., Ahn, E., Zhang, M., and Liu, H. (2023)" without any modification or reference added. Additionally, the resolution of these images is not clear enough for understanding.

4. PeerJ Computer Science does not specify a particular reference format, but, authors should aim for consistency and clarity. If the manuscript follows APA style, citations should be in the format of (Author, Year). Please verify the citation format, such as in line 130 ("To increase the accuracy of object detection, it combines anchor boxes, anchor box scaling, and anchor box ratios Yang et al. (2022)"). Check other citations for consistency. For APA reference style, please try to cite references in the text by name and year in parentheses. For examples:
a)Negotiation research spans many disciplines (Thompson 1990).
b) This result was later contradicted by Becker and Seligman (1996).
c) This effect has been widely studied (Abbott 1991; Barakat et al. 1995a, b; Kelso and Smith 1998; Medvec et al. 1999, 2000).

5. PeerJ Computer Science suggests standard manuscript sections such as Background, Results, Discussion, and Conclusions. It would be beneficial to add a separate discussion section after the results where the limitations of the model/experiment can be addressed.
6. Could you please clarify whether the reported results in lines 29 and 30 of the abstract pertain to tooth detection or the detection of four types of radiographic bone loss (RBL)?
7. In Table 1, all the referenced articles do not focus their research on Periodontitis Bone Loss Diagnosis as you have mentioned. For instance, "Celik, M. E. (2022)" focused on mandibular 3rd molar tooth impaction detection, and Astuti et al.'s research concentrated on tooth detection. Please revise the referenced articles in the table to focus on topics related to your research. Additionally, providing a descriptive analysis of previous research findings and limitations addressed in the current study would enhance the Introduction section's relevance. Also, be specific and explicit about the research objectives.

Experimental design

8. In line 177, you mentioned “training your model for 100 epochs with a batch size of 32”, could you provide details about other hyperparameters? Did you use a pretrained model? Please explicitly state the image size to understand the compatibility with your experimental environment.

9. Were Figures 4 to 9 derived from training your proposed model? Did you utilize a validation set? Additionally, the captions of these figures are not explicit.

10. In line 162, you defined the F1 score as "Model performance is inversely correlated with the F1-score." Could you please explain how your model's performance is inversely related to this metric?

11. In lines 212 and 213, you mentioned "Table 5 shows the experimental results of the proposed YOLOv7 model in detecting periodontitis bone loss in panoramic radiographs." Did you mean your modified YOLOv7 model? How many images did you utilize to evaluate your model? The title of Table 5 seems irrelevant.
12. In section 4.2, you have shown a comparative analysis between different YOLO series and your proposed YOLOv7-M, which is modified from YOLOv7. YOLOv7 has several versions (YOLOv7s, YOLOv7l,…). Which architecture did you utilize for your modified model and original YOLOv7 model? Did you utilize the same types of augmentation, loss function and hyperparameter settings? Could you report the inference time of the implemented architectures as well? Your research utilized the dataset of Zhengmin Kong et al.’s research. A comparison with their outcome is expected.

Validity of the findings

13. You have summarized your key contributions as follows: “The focus module improves the model’s ability to capture fine-grained details by emphasizing relevant regions of the input. It achieves this by reducing the spatial dimensions of the feature maps while preserving their essential features.” Did you verify the effects of these modules individually? Which one of the newly added modules is more significant to improve the performance?

Additional comments

It appears that the manuscript was not meticulously prepared. The narrative seems overly extensive and could be streamlined to improve clarity and effectiveness. There is a lack of cohesion throughout the article, and it would benefit from proper elucidation across sections. Additionally, the key contributions highlighted in lines 77-84 require further reflection. Language refinement is necessary to ensure clear understanding by an international audience.

Cite this review as

---

## Round 0.2 · Major Revisions

Dear authors,

I have taken over handling this submission because the original Academic Editor is not available.

You are advised to critically respond to all comments point by point when preparing an updated version of the manuscript and while preparing for the rebuttal letter. Please address all comments/suggestions provided by reviewers, considering that these should be added to the new version of the manuscript.

Kind regards,
PCoelho

Reviewer 2 ·

Basic reporting

At first, I would like to express my appreciation for the authors' efforts and hard work in improving their manuscript based on the comments from the first review. However, there are still some inconsistencies that need to be addressed:
1. Figure 11: The ground truth and predicted images for all reported model outcomes are not consistent. The authors should clearly indicate which images correspond to the outcomes of their implemented algorithms. Additionally, the provided code is based on the YOLOv7 GitHub repository baseline, making it difficult to understand their implemented models.
2. Inconsistencies in reporting results:
In the abstract, you mention:
“The proposed YOLOv7-M model was evaluated on a tooth detection dataset and demonstrated superior performance, achieving an F1 score, precision, recall, and mAP of 92.5, 91.7, 87.1, and 91.0, respectively.”
In the discussion section, you state:
“The precision, recall, and F1 scores of 91.7%, 87.1%, and 93.5%, respectively, along with an mAP of 93.0%, highlight the model’s robustness and reliability in detecting periodontitis.”
These discrepancies need to be resolved to ensure consistency in the reported results.

Experimental design

3. Validation experiment:

Since the primary contribution of your work lies in reporting results for the modified YOLOv7-M model compared to other studies, you should confirm your findings with a cross-validation experiment. Stratified 5-fold cross-validation is suggested. The current random data-splitting process does not adequately ensure superiority over the baseline work of Zhengmin Kong et al.

Validity of the findings

4. Comment 12:
There are no reflections on comment 12 in the revised manuscript. Please refer to the article for reporting the methods:
https://doi.org/10.1016/j.jdent.2021.103610.

Cite this review as

---

## Round 0.3 · Minor Revisions

Dear authors,

Thanks a lot for your efforts to improve the manuscript.

Nevertheless, some concerns are still remaining that need to be addressed.

Like before, you are advised to critically respond to the remaining comments point by point when preparing a new version of the manuscript and while preparing for the rebuttal letter.

Kind regards,
PCoelho

Reviewer 2 ·

Basic reporting

I appreciate your hard work. It has been an honor to review your paper, and I would like to extend my sincere congratulations.

Experimental design

New experiments show the robustness of the proposed methods.

Validity of the findings

The new five-fold experiments demonstrate results consistent with the previous random data splitting, further confirming the robustness of the proposed model for periodontal bone loss detection.

Additional comments

I kindly request a more careful check of the manuscript before final submission. For example, in line 321, the statement "YOLO-v7-M achieves the highest mAP score of 0.930" may contain a typographical error. There is also room to improve the clarity of the table title—consider revising it to something like "Table 5. RBL Annotations for Teeth in the Dataset."

Furthermore, some descriptions appear to be repeated or placed in unrelated sections, such as the Discussion and Conclusion. Other sections, like the Methods, could benefit from more detailed explanations. I encourage you to address these issues before your final submission to ensure that the paper is coherent, concise, and more explicit.

Cite this review as

---

## Round 0.4 · accepted · Accept

Dear authors, we are pleased to verify that you meet the reviewer's valuable feedback to improve your research.

Thank you for considering PeerJ Computer Science and submitting your work.

Kind regards
PCoelho